# Trends in Physiotherapy of Chronic Low Back Pain Research: Knowledge Synthesis Based on Bibliometric Analysis

**DOI:** 10.3390/healthcare12161676

**Published:** 2024-08-22

**Authors:** Urška Šajnović, Peter Kokol, Jernej Završnik, Helena Blažun Vošner

**Affiliations:** 1Community Healthcare Center dr. Adolf Drolc Maribor, 2000 Maribor, Slovenia; peter.kokol@um.si (P.K.); jernej.zavrsnik@zd-mb.si (J.Z.); helena.blazun@zd-mb.si (H.B.V.); 2ECM Maribor, Alma Mater Europaea University, 2000 Maribor, Slovenia; 3Faculty of Electrical Engineering and Computer Sciences, University of Maribor, 2000 Maribor, Slovenia

**Keywords:** physiotherapy, chronic low back pain, knowledge synthesis, bibliometric analysis

## Abstract

Background: Physiotherapy and chronic low back pain (CLBP) form a broad and quickly developing research area. The aim of this article was to holistically, thematically and chronologically analyze and synthesize the literature production in this research area and identify the most prolific research entities and research themes. Methods: This article quantitatively and qualitatively analyzed research literature production harvested from the Scopus bibliometric database, using a triangulation of bibliometric and thematic analysis. For this, Excel 2024, Bibliometrix Biblioshiny 4.1 and VOSviewer version 1.6.20 softwares were used. Results: In the Scopus database, 2843 data sources were found, which were published between 1974 and 26 February 2024. The growth trend has been linearly positive since the beginning of publication, and after 2018 exponential growth began. A review of the most prolific entities showed that the most literature was published in America, Europe and Australasia. The thematic analysis of the information sources identified six main themes (pathophysiology of CLBP and the quantification assessment tools, diagnostics and CLBP treatment, CLBP questionnaires and surveys, quality of life, complementary methods in physiotherapy and psychosocioeconomic aspects), while the chronological analysis revealed three main areas of development: assessment tools, CLBP processing and study methodology. Conclusions: The results of this bibliometric study present a good starting point for further research, providing taxonomy and research landscapes as a holistic framework offering multidisciplinary knowledge about CLBP, while chronological analysis provides a basis for identifying prospective research trends. This article offers an interdisciplinary view of the current issue of public health. The results of this study provide a basis for the development of both the physiotherapy and epidemiological fields.

## 1. Introduction

Chronic low back pain (CLBP) is the leading cause of disability globally and the greatest cause of years lived with disability (YLD) in the world, which affected 619 million individuals globally in 2020 [1,2,3]. Physiotherapy has an enormous role in the rehabilitation of CLBP. Over time, various research topics have emerged that deal with CLBP and its connection with physiotherapy. A multidisciplinary approach to rehabilitation plays an increasingly important role, as other health sciences, such as kinesiology, psychotherapy, psychology, etc., are increasingly joining the physiotherapy treatment of CLBP [4,5,6,7,8,9,10]. Consequently, physiotherapy and CLBP represent a broad science field, resulting in a large and increasing number of research publications.

Bibliometrics is a synthesis method that enables the analysis of large corpora of publications on a macroscopic and microscopic level, and it has become a popular approach in medical research [11]. Its main advantage is that it is discipline-independent and that, contrary to traditional reviews where the analysis is performed on a small sample of selected publications, bibliometric analysis is performed on a whole corpus of publications. However, previous bibliometric reviews have explored the association between physiotherapy (mainly exercise effects) and CLBP on a microscopic level only. Additionally, socioeconomic and interpersonal aspects (patient–physiotherapist) have not been holistically explored. The studies primarily show geospatial [12,13] and temporal landscapes [12,13,14,15] and cluster analysis of co-authorships and keywords [13,14,15,16].

To fill this research gap, we used a mixed-method approach called Synthetic Knowledge Synthesis (SKS) [11]. SKS combines bibliometrics, bibliometric mapping and thematic analysis to perform quantitative descriptive bibliometrics (recognizing the most prolific source titles, institutions, countries and funding institutions) and to qualitatively identify the most popular research topics, current trends and research gaps in the field of physiotherapy employment in CLBP. Since these are two independent research fields, we analyzed the research landscapes and, within them, the associations/relations of several research terms and categories. In such a way, we managed to identify overlapping research topics and themes and the timeline of their progress, thus providing an interdisciplinary but holistic view of the research field. For this purpose, we formulated the following research questions:What is the volume, scope and dynamics of the research literature production on physiotherapy in CLBP?What is the spatial distribution of physiotherapy in CLBP literature production among the most prolific source titles, institutions, countries and funding institutions?What is the most prolific physiotherapy in CLBP research themes, how have they evolved over time and what is their association?

The main objective of our study was to provide a framework that can support the research community in solving the theoretical and practical challenges faced when using physiotherapy in CLBP. Researchers and health professionals can use the framework to improve their understanding of the area and to identify further knowledge development directions. The framework can also inform novice researchers, administrators and patients without specific knowledge to develop a perspective on the most important research and health practice dimensions. Finally, the framework can provide a starting point for more formal knowledge syntheses approaches like systematic reviews and meta-analyses.

## 2. Materials and Methods

### 2.1. Search Procedure

Our search was conducted on 26 February 2024. Data were obtained from Scopus (Scopus, Elsevier, Amsterdam, The Netherlands). The Scopus database was chosen because it contains peer-reviewed articles, includes the Pubmed database and offers simultaneous retrieval of a large amount of data. The search (articles, reviews, conference papers, notes, letters, short surveys, book chapters, editorials and errata) was limited to the period between 1974 (when the first publication was indexed in Scopus) and 2024. We also included the year 2024 because we wanted to analyze the latest research trends and describe how they impact on the topic of older research. Sources were limited to literature written in English. Within this study, the search string ‘chronic low back pain’ was used (AND ‘physiotherapy’ OR ‘trust’ OR ‘quality of life’ OR ‘stress’) in the titles, abstracts and keywords. No further inclusion or exclusion criteria were used.

### 2.2. Data Analysis

The metadata of the corpus of harvested publications were extracted and exported to Vosviewer (Leiden University, The Netherlands) [17], Bibliometrix [18] and Excel softwares (Microsoft Office, Microsoft Corporation, Redmond, WA, USA). The metadata were pre-processed with Excel’s built-in functions, which enabled tabulation, data cleansing and the creation of tables and graphs. Only complete abstracts were exported to VOSviewer for bibliometric mapping of authors’ keywords. We used Bibliometrix software [18] to perform a descriptive analysis regarding the number of scientific journals, institutions, countries, authors and authors’ keywords.

### 2.3. Thematic Analysis

Thematic analysis was performed on bibliometric maps. A bibliometric map is a network of nodes: in our case, authors’ keywords. Links in the network denote various relations between keywords, proximity similarity between keywords and the node size and popularity of a keyword. A cluster is a group of thematically strongly associated keywords. Author keywords in a single cluster were used as meaningful units of information for the thematic analysis. The unit’s node size, links and proximity between units in single clusters were analyzed first to form categories. Next, categories were analyzed to identify the cluster theme.

## 3. Results

### 3.1. Volume and Prevalence of the Research Literature Production

A search in the Scopus database identified 2843 information sources (among them, there were 2281 articles, 403 reviews, 38 conference papers, 34 notes, 25 letters, 25 short surveys, 16 book chapters, 14 editorials and 7 errata). They were published between 1974 and 2024. All sources of information were included in this analysis with the aim of obtaining the most objective and broad view of the research area.

Literature sources were published in 80 different countries, one of which was not identified. Most publications were issued in the USA (753), followed by the United Kingdom (265), Germany (238), Australia (225) and the Netherlands (188). The most prolific countries came from five continents (North America, Europe, Australia, South America and Asia), but otherwise there were information sources published on all continents, which demonstrates the importance of the research area worldwide. All the most productive countries have good economies and advanced health systems. We also recognized the most productive institutions of the 860 source titles: the University of Sydney (77), Australia; Vrije Universiteit Amsterdam (56), Netherlands; Universiteit van Amsterdam (52), Netherlands; Universiteit Maastricht (52), Netherlands; Harvard Medical School (47), USA; EMGO Institute for Health and Care Research (46), Netherlands; and VA Medical Center (46), USA. The analysis showed that 10,692 authors participated in the writing of the bibliography, of which 167 were single-authored docs, while the rest involved at least two or more authors (an average of 5.45 co-authors per doc). Almost a fifth of the authors (19.64%) participated in international co-authorship, which indicates a relatively high level of international cooperation in research.

The analysis showed that 937 out of a total of 2843 information sources were sponsored, representing almost one-third of all sources. The institutions that provided the most funding, which sponsored the publication of research data, are represented in Table 1. These are mostly nationally funded institutes and organizations.

Information sources in the research area of physiotherapy of CLBP were published in 860 source titles. There was a total of 2846 information sources. The most prolific journals, which published the most productive information sources, are presented in Table 2. These represent one-third of all resources (30.75%). As can be seen from the table, journals are coming mostly from the spine and pain categories.

### 3.2. The Dynamics of the Research Literature Production

The trend of research literature production is presented in Figure 1. For better resolution, information sources are shown every two publication years. The year 2024 is also included in the graph to show that 28 articles were already accessible in the Scopus database in the first two months alone. The first information sources were published in 1974. In the years 1973 to 1992, production was stable, and up to 10 sources were released per year on average. After that year, linear growth began with a peak in 2004, when the production reached 68 articles annually. The trend of linear growth continued with a peak of 131 articles in 2016. In the meantime, a slight drop can be seen in 2018, but the number began to increase exponentially after that year (from 179 up). From 2021 onwards, the number of published literatures has risen to over 200 articles per year, and the trend of further exponential growth is noticeable. This coincides with the growing trend of CLBP [1,3,19] and the development of physiotherapy and other interdisciplinary health sciences [20,21]. Our analysis shows a 6.89% annual growth rate in scientific literature in the research field.

The graph also shows the growth dynamics of individual genres of scientific literature: conference papers, reviews and original articles. As can be seen from the graph, the trend of original articles most closely follows the dynamics of the entire scientific literature. The first increase in publications can be seen after 2014, and the second, larger one, after 2018. After 2020, the number exceeded 200 publications per year.

Conference articles are hardly detectable, which does not mean that they do not exist, but that conference proceedings from this research field are not indexed in Scopus. Abstracts of conference papers are published in journals.

The reviews had a steady dynamic trend; the peak can be seen in 2008, and after 2020 their number rose above 22 publications annually (up to 22 publications annually by 2018). The peak in 2008 actually represents an artifact, as in this year exceptionally more articles (23) were published than usual (up to a maximum of 16).

The growth trend of literature from 2018 onwards is shown in Figure 2. A strong growth in the production of literature is evident.

### 3.3. The Thematic Analysis

As the interpretation of the bibliometric method is completely subjective, connections between different topics/keywords were searched. Clusters were identified and subsequently named according to the expertise of the authors in each research field. Six clusters were induced using the SKS: pathophysiology of CLBP and the quantification assessment tools (yellow), diagnostics and CLBP treatment (red), CLBP questionnaires and surveys (violet), quality of life (light blue), complementary methods in physiotherapy (dark blue) and psychosocioeconomic aspects (green). Clusters are also shown in the scientific landscape on Figure 3. Clusters were named based on the analysis of the most representative keywords and links between them. Due to the complexity of the themes, the categories were formed and then joined into topics, as shown in Table 3. Based on the most prolific keywords, categories and themes, three-to-five of the most recent and most cited articles for each category were identified, analyzed and summarized.

### 3.4. The Chronological Analysis

The chronological development of research terms in the literature on CLBP and physiotherapy is illustrated in Figure 4. Chronological analysis showed the following main time periods: the first period up to 2010, second period between 2010 and 2013, third period between 2013 and 2014, fourth period between 2015 and 2016, fifth period between 2016 and 2018 and the sixth period from 2018 onwards. Early research until 2010 (dark blue) examined CLBP as a chronic disease [46,47,48,49,50,51,52], which was studied with follow-up methodological articles. They developed scoring and rating systems for the numerical identification of pain. In the interim period until 2013 (medium blue), research focused on disease severity [53]. Later, between 2013 and 2014 (blue–green), physiotherapy and its therapeutic modalities, especially exercise and kinesiotherapy [54,55,56,57,58,59], began to appear more prominently in the literature. At the same time, articles began to appear on pain-relieving therapy and analgesia with NSAIDs [60]. It is important to note that in this period younger adults began to be mentioned as the target population [61], with which studies, consistent with the pathophysiology of CLBP, began to recognize other risk factors for CLBP, not just degenerative ones. In the next period, between 2015 and 2016 (yellow), studies focused on the duration of CLBP and quality of life [5,61,62,63,64], thus broadening the picture of the causative factors for the occurrence of CLBP. At the same time, the role of the physiotherapist, especially his approach, as a key factor in the rehabilitation of CLBP began to be mentioned more significantly [65,66,67,68]. In addition, anxiety began to be included in the assessment of the functional status of a patient with CLBP [69,70,71,72,73,74], which was a predisposition for further research in the field of psychology and the formation of interdisciplinary links between medicine (CLBP), physiotherapy, psychology and psychotherapy. Between 2016 and 2018 (orange and red), psychology began to play a more prominent role, which influenced the development of a multidisciplinary approach to the treatment of CLBP. In addition to physiotherapy, psychotherapy, behavioral–cognitive therapy and associated branches of treatment began to appear in the rehabilitation of CLBP [5,7,8,75]. Newer research since 2018 mostly refers to cohort analyses using newer clinical assessment tools (such as the Numerical Pain Rating System, Oswestry Disability Index and Rolland–Morris questionnaire) [76,77,78] with the aim of developing the best possible clinical outcome of CLBP rehabilitation and treatment. Newer studies also show the effectiveness of high-intensity laser therapy, which also has good effects in the rehabilitation of CLBP [79,80,81,82].

Based on a detailed chronological analysis, we analyzed the following chronological developments:Development of pain assessment tools: *pain measurement → pain assessment → pain management*. These tools first measure and evaluate pain, in order to ultimately contribute to the most functional treatment possible;Development of CLBP processing: *disease severity → disability evaluation → disease duration → treatment duration → clinical outcomes and effectiveness*. The evolution of CLBP processing first began with assessing the severity and then the inability to perform normal bodily functions caused by the pain. The latter affected the duration of the disease, which in turn resulted in the duration of the treatment. Research in this area has continued in the field of studying the clinical outcomes and effectiveness of therapeutic methods of treating CLBP;Development of study methodology: *follow-up studies → comparative studies → randomized control trials → meta-analysis → cohort analysis*. The methodological design first started with follow-up studies, followed by comparative studies and later by randomized control trials. Based on the conclusions of these studies, after further research authors began to summarize the main findings and suggest further possibilities for the development of physiotherapy for CLBP in meta-analysis. They were followed by cohort analyses studying causal factors over time.

## 4. Discussion

The aim of our article was to qualitatively analyze the literary production in the field of physiotherapy for CLBP using the bibliometric method and bibliometric mapping. The present article has also some limitations; firstly, the interpretation of the bibliometric term map clusters and term association analysis (bibliometric mapping) is qualitative and consequently subjective. Additionally, the fact that we only used a Scopus database means that we might have overlooked some relevant literature from the analysis. Another limitation is the selection of only English articles; therefore, we might also lose some important resources in other languages. However, our study also has a strength since bibliometric analysis revealed several characteristics of the research literature production, as well as the important institutions, journals and the research themes in physiotherapy in CLBP, which could help healthcare professionals and others interested in their future research activities. Additionally, as far as we know, bibliometric analysis is not used in the analysis of this research field yet, which represents another strength of this article.

Previous studies show that the method of bibliometric analysis with a combination of bibliometric mapping and thematic analysis is a useful method for analyzing a large amount of literary production [20,83,84,85,86]. This methodology allowed us to place our research field in a broader interdisciplinary context with the aim of understanding the diversity of research disciplines and connecting concepts, methods and conclusions between them.

This study shows a positive linear trend of increasing research literature since 1974, with exponential growth after 2019, when the publication of over 200 articles per year began. There is also a positive linear trend of increasing research literature since 1974, with exponential growth after 2019, when the publication of over 200 articles per year began. This phenomenon coincides with an increase in the prevalence of CLBP [43,45,87,88,89] and the number of sick leave days and years lived disabled (YLD) [1,90,91]. It is expected to continue to rise sharply, in line with the predictions of studies on the increase in psychophysical disability and the amount of costs due to CLBP. Because CLBP is a global problem, it is being addressed by researchers worldwide. Our analysis also confirmed the publication of data on all continents, of which North America and Europe are leading. The reason for this is probably also the fact that America has the largest and most diverse healthcare system in the world [83]. In addition, unemployment and the sick leaves of workers represent a strong deficit in the economic aspect of the most industrially developed countries.

Thematic analysis of data information yielded a scientific map with six main thematic categories: CLBP pathophysiology and assessment tools, CLBP diagnosis and treatment, CLBP questionnaires and surveys, quality of life, complementary methods in physiotherapy and psychosocioeconomic aspects. It is obvious that their subcategories are interrelated, even though they come from different thematic areas, and some of them even duplicate within different clusters. The latter explains the development of the scientific literature on the multidisciplinary approach to the physiotherapy treatment of CLBP, which began to develop more strongly after 2010 [4,5,8,10,92,93,94]. The most prolific publications are *Spine*, *BMC Musculoskeletal Disorders*, *European Spine Journal*, *Spine Journal* and *Pain*, which have relatively high impact factors (IF; from 2.3 to 7.9).

It is interesting to note the development of the data literature over time. From the chronological scientific map, it can be seen that until 2013 the follow-up and comparative studies focused on classical CLBP physiotherapy. After 2016 these were followed by meta-analytic and systematic reviews studies [7,95,96]. These began to consider a multidisciplinary approach to CLBP rehabilitation and to study classical physiotherapy with associated treatment methods (psychotherapy, psychology, kinesiology, balneotherapy, yoga, etc.). Early research focused primarily on the neurological association with CLBP. Later, after 1980, researchers also began to mention the psychological aspect of pain and its influence on rehabilitation. The growth of data sources in this area, especially after 1990, added to the broader picture of pain implicit theories, which included pain scoping, depression, anxiety and catastrophizing [97,98,99,100,101]. It seems that the body’s level of connection with CLBP is expanding simultaneously with multidisciplinary approaches. After 2010, intensive research also began in the field of immunology [30,102,103,104], thus extending the neurological connection to the psychoneuroimmunological connection.

Despite the increasing emphasis on combining classical physiotherapy and related health sciences (kinesiology, psychology and psychotherapy), it seems that the basic approach of the physiotherapy process remains the basis for successful rehabilitation of CLBP. The physical therapist’s verbal and non-verbal relationship with the patient forms the basis of trust from which the effects of the therapy are directly and indirectly reflected [41,65,105,106].

As already mentioned, previous bibliometric reviews [12,13,14,15,16] did not focus in detail on the socioeconomic aspect and interpersonal communication between a patient with CLBP and a physiotherapist. Only Zheng [13] partially mentioned the importance of contacting primary care providers, as they are always the point of first contact for these patients. Based on the comparison of the aforementioned reviews and our study, we can draw parallels regarding the dynamics of the production of research literature and the analysis of the most prolific countries, institutions and scientific journals. Despite the more detailed thematic evolution [12,14], none of the studies extracted similar chronological developments to our analysis (development of pain assessment tools, CLBP processing and study methodology), which could also be a consequence of the qualitative interpretation of the chronological analysis.

The present study brings interesting findings, which can be an excellent basis for further research in this field, especially in the direction of developing multidisciplinarity in the treatment of CLBP. The findings of the present article greatly contribute to the development of science in the field of physiotherapy and CLBP, as they show the thematic and chronological analysis and the dynamics of the creation, scope and distribution of scientific literature in this field. The landscapes in the present study can help the research community to solve the theoretical and practical challenges of using artificial intelligence in the fields of public health, physiotherapy, pain therapy and occupational medicine.

Researchers and practitioners can use the findings of this study to improve their understanding of the research area. In addition, the results can serve as a catalyst for the further development of knowledge in this area. This study is not only interesting for experienced researchers, but also for novice researchers, research managers, medical students, directors of health institutions and government decision makers, as it helps them develop a broader view of the research topic and the connection between important segments of this broad public health problem. Last but not least, the landscape can also serve as a starting point for further systematic studies that synthesize the findings so far (such as meta-analyses and systematic reviews).

## 5. Conclusions

Our bibliometric study has shown that CLBP and physiotherapy form a broad interdisciplinary scientific research field supported by studies from all over the world. A taxonomy, a chronological review and an extract of the most productive entities set the frame to the research field of physiotherapy in CLBP. This could be useful for further research, and also provides new perspectives on the existing state of the scientific literature in this area. Bibliometric analysis and mapping provide a basis for thematic analysis, which is an effective tool for analyzing research literature production. This study offers six main themes emerging from the interdisciplinary field of physiotherapy and CLBP and provides a basis for further integrative examination of research issues in this field.

## Figures and Tables

**Figure 1 healthcare-12-01676-f001:**
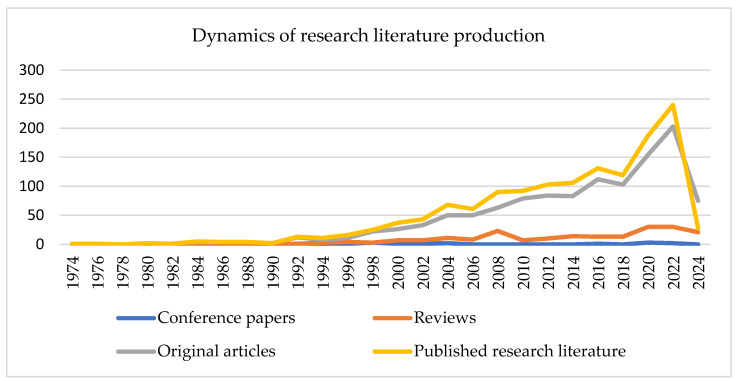
The dynamics of research literature production.

**Figure 2 healthcare-12-01676-f002:**
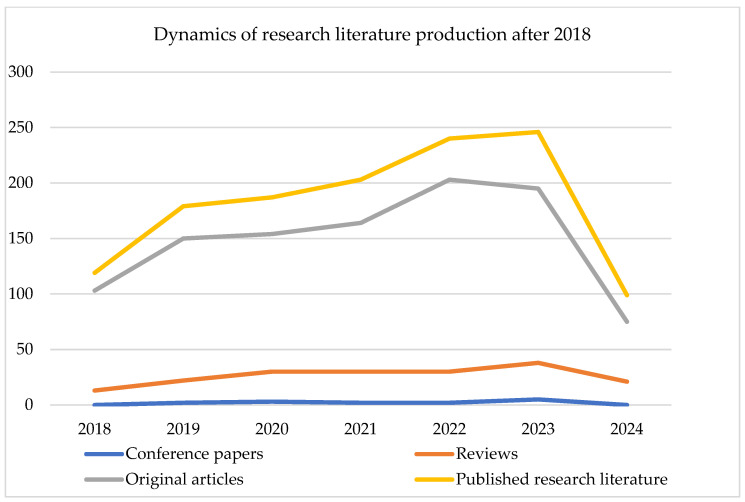
The dynamics of research literature production after 2018.

**Figure 3 healthcare-12-01676-f003:**
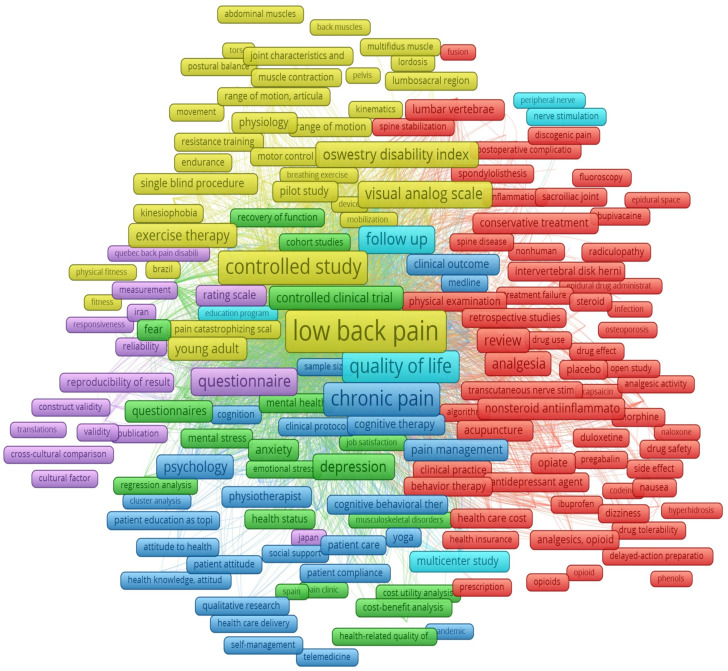
The bibliometric map of the research on the use of physiotherapy in CLBP.

**Figure 4 healthcare-12-01676-f004:**
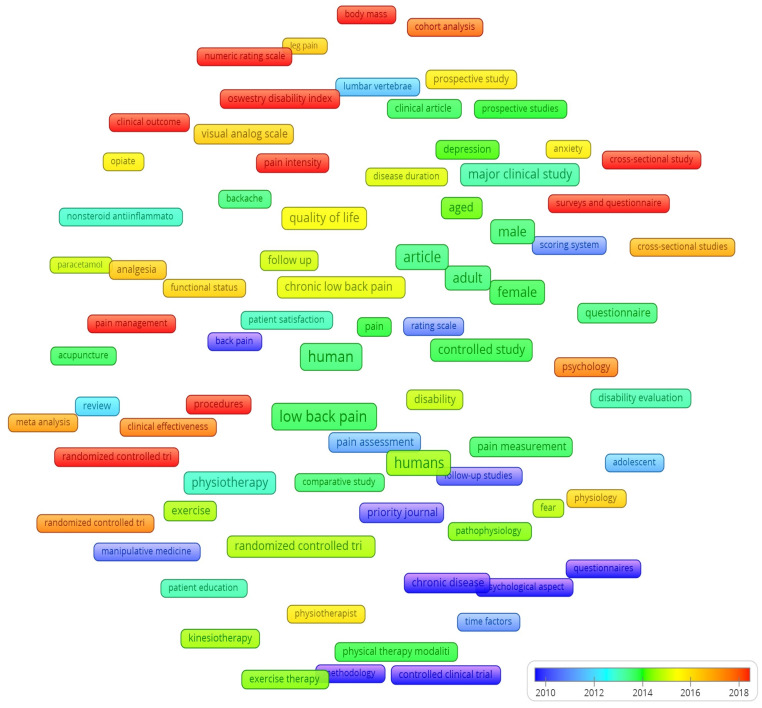
Chronological diagram of the research topics.

**Table 1 healthcare-12-01676-t001:** List of the most prolific institutions.

Institution	Number of Funded Publications	Percentage of Funded Information Sources (in %)
National Institutes of Health	84	8.97
National Center for Complementary and Integrative Health	67	7.15
Coordenação de Aperfeiçoamento de Pessoal de Nível Superior	30	3.20
National Health and Medical Research Council	28	2.99
Fundação de Amparo à Pesquisa do Estado de São Paulo	23	2.46
National Natural Science Foundation of China	23	2.46
Pfizer	18	1.92
U.S. Department of Veterans Affairs	18	1.92
National Institute on Aging	17	1.81
National Institute of Child Health and Human Development	16	1.71

**Table 2 healthcare-12-01676-t002:** List of the most prolific journals and their bibliometric indicators.

Source Title	Number of Sources	Percentage of All Sources (in %)	IF 2022 ^1^	H-Index of Journal ^2^	Quartiles (Q) ^3^
Spine	129	6.79	3.0	292	Neurology (clinical) (Q1) Orthopedics and Sports Medicine (Q1) Sports Science (Q1)
BMC Musculoskeletal Disorders	81	4.27	2.3	122	Orthopedics and Sports Medicine (Q2) Rheumatology (Q2)
European Spine Journal	52	2.74	2.8	164	Orthopedics and Sports Medicine (Q1) Surgery (Q1)
Spine Journal	52	2.74	4.5	136	Neurology (clinical) (Q1) Orthopedics and Sports Medicine (Q1) Surgery (Q1)
Pain	51	2.69	7.4	296	Anesthesiology and Pain Medicine (Q1) Neurology (Q1) Neurology (clinical) (Q1) Pharmacology (Q1)
Journal Of Back and Musculoskeletal Rehabilitation	48	2.53	1.6	39	Orthopedics and Sports Medicine (Q3) Physical Therapy, Sports Therapy and Rehabilitation (Q2) Rehabilitation (Q2)
Clinical Journal of Pain	47	2.48	2.9	145	Anesthesiology and Pain Medicine (Q1) Neurology (clinical) (Q2)
Pain Medicine United States	46	2.42	No data	No data	No data
Journal of Pain Research	44	2.32	2.7	71	Anesthesiology and Pain Medicine (Q2)
BMJ Open	34	1.79	2.9	160	Medicine (miscellaneous) (Q1)

Notes: ^1^ IF according to the Bioxbio Journal impact search engine. ^2^ H-index according to the SJR (Scimago Journal and Country Ranks). ^3^ Quartiles (Q) according to the SJR (Scimago Journal and Country Ranks).

**Table 3 healthcare-12-01676-t003:** Themes/categories related to research found in the newest/most cited papers and their focuses of research.

Theme (Color)	Representative Keywords	Categories	The Newest/Most Cited Papers	Focus of Research
Pathophysiology of CLBP and the assessment tools (yellow)	Oswestry Disability Index, pilot studies, single-blind method, exercise therapy, pathophysiology, range of motion, spine mobility	Pain quantification tools, Muscle physiology, Rehabilitation, Kinesiology	- Visual analogue scale, numeric pain rating scale and the McGill pain Questionnaire: an overview of psychometric properties [22] - Studies Comparing Numerical Rating Scales, Verbal Rating Scales, and Visual Analogue Scales for Assessment of Pain Intensity in Adults: A Systematic Literature Review [23] - Pain biology education and exercise classes compared to pain biology education alone for individuals with chronic low back pain: A pilot randomised controlled trial [24] - Physiotherapeutic treatment associated with neuroscience of pain education for patients with chronic non-specific low back pain—single-blind randomized pilot clinical trial [25] - Biopsychosocial Rehabilitation on Short-Term Pain and Disability in Chronic Low Back Pain: A Systematic Review with Network Meta-Analysis [10]	The Visual Analogue Scale (VAS) and the Numeric Pain Rating Scale (NPRS) are the most commonly used scales of perceived pain intensity, but otherwise there is no established gold standard for pain measurement in the literature. Physiotherapy interventions through behavioral (teaching the patient about the biology of their pain) or instructional modalities demonstrate better treatment outcomes for CLBP.
Diagnostics and CLBP treatment (red)	Magnetic resonance imaging (MRI), diagnostic imaging, conservative treatment, risk assessment, radiculopathy, herniation, operation	Diagnostics, Analgesia, Risk assessment, Pathology, Other treatment methods	- Ultra-short echo time MR imaging in assessing cartilage endplate damage and relationship between its lesion and disc degeneration for chronic low back pain patients [26] - The enhanced connectivity between the frontoparietal, somatomotor network and thalamus as the most significant network changes of chronic low back pain [27] - The Longitudinal Relationship Between Emotion Regulation and Pain-Related Outcomes: Results From a Large, Online Prospective Study [28] - Fear-avoidance and its consequences in chronic musculoskeletal pain: a state of the art [29] - Role of CD14-positive cells in inflammatory cytokine and pain-related molecule expression in human degenerated intervertebral discs [30]	MRI is the most widespread method for researching pain networks in the brain. A newer method is ultra-short echo time (UTE) MRI. Research indirectly suggests a psychoneuroimmunological connection in CLBP. Pain-related fear and avoidance appear to be essential features for the development of chronic pain. The thalamus plays the most important role in regulation of pain-related emotions. Expression of pain-related molecules is mediated by CD14+ cells via inflammatory cytokines.
CLBP questionnaires and surveys (violet)	Reproducibility, validity, assessment, rating scale, catastrophizing, nerve blockade	Questionnaires and surveys, Pain, Patients’ impact	- Evaluating Common Outcomes for Measuring Treatment Success for Chronic Low Back Pain [31] - Roland-Morris Disability Questionnaire and Oswestry Disability Index: Which Has Better Measurement Properties for Measuring Physical Functioning in Nonspecific Low Back Pain? Systematic Review and Meta-Analysis [32] - Understanding the link between depression and pain [33] - Coping Styles, Pain Expressiveness, and Implicit Theories of Chronic Pain [34]	Studies recommend measuring several outcomes to assess the strength of CLBP and the effectiveness of physiotherapy, namely functional (Oswestry Disability Index, Rolland–Morris Disability Index, etc.), pain (NPRS, Pain Disability Index, etc.), psychosocial (Fear Avoidance Beliefs Questionnaire) and other (return to work, complications or adverse effects, etc.) outcomes and quality-of-life assessment (SF-36, etc.). Negative behavioral emotion regulation is reported to result in spiraling negative affect and subsequent CLBP relapse.
Quality of life (light blue)	Sleep quality, quality of life, functional status, treatment outcome, neuromodulation, balneotherapy	Quality of life, Intermethod comparison, Newer therapeutic methods	- Assessment of depression, anxiety, sleep disturbance, and quality of life in patients with chronic low back pain in Korea [35] - Effects of Lifestyle Interventions on the Improvement of Chronic Non-Specific Low Back Pain: A Systematic Review and Network Meta-Analysis [36] - Multidisciplinary biopsychosocial rehabilitation for chronic low back pain: Cochrane systematic review and meta-analysis [5] - Exposure to greenspaces could reduce the high global burden of pain [37]	Depression, anxiety, coping behavior and catastrophizing are the main components of the quality-of-life assessment and influence the strength of CLBP and the success of physiotherapy. The concept of a multidisciplinary approach emphasizes the connection between the psychological and physical components of pain. More recent studies are also focused on investigating the effects of the environment and report that a calmer and green living environment should have a positive effect on the expression of CLBP and pain in general.
Complementary methods in physiotherapy (dark blue)	Physical therapy modalities, clinical protocol, yoga, cognitive behavioral therapy, coping behavior, interpersonal communication, mindfulness, lifestyle	Physiotherapy, Complementary therapies, Personal relationships, Coping behavior	- Effect of multidimensional physiotherapy on non-specific chronic low back pain: a randomized controlled trial [38] - Meditation-Based Therapy for Chronic Low Back Pain Management: A Systematic Review and Meta-Analysis of Randomized Controlled Trials [39] - The Influence of the Therapist-Patient Relationship on Treatment Outcome in Physical Rehabilitation: A Systematic Review [40] - The role and function of body communication in physiotherapy practice: A qualitative thematic synthesis [41] - Cognitive and emotional control of pain and its disruption in chronic pain [42]	The physiotherapist–patient relationship plays an important role in the success of physiotherapy, as it affects the patient’s trust. The effectiveness of rehabilitation is also influenced by non-verbal communication, the patient’s ability to cope with pain and associated relaxation methods (meditation).
Psychosocioeconomic aspects (green)	Cost–benefit analysis, healthcare costs, sick leave, social aspects, psychotherapy, depression, prognosis	Economy, Absenteeism, Socioeconomic factors, Disability, Risk factors	- The Epidemiology and Economic Consequences of Pain [43] - Effects of an early multidisciplinary intervention on sickness absence in patients with persistent low back pain—a randomized controlled trial [44] - Global, regional, and national incidence, prevalence, and years lived with disability for 310 diseases and injuries, 1990–2015: a systematic analysis for the Global Burden of Disease Study 2015 [45]	Research shows an increasing prevalence of CLBP and associated healthcare costs. Early multidisciplinary treatment of CLBP is essential to reduce the rate of sick leave and work disability.

## Data Availability

Data sharing is not applicable. No new data were created or analyzed in this study.

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
