# Peer review of "Trends in Physiotherapy of Chronic Low Back Pain Research: Knowledge Synthesis Based on Bibliometric Analysis"

_healthcare, 2024, doi:10.3390/healthcare12161676_

Round 1

Reviewer 1 Report

Comments and Suggestions for Authors 1. The English should be fluent and clear, with smooth transitions between paragraphs. Correct spelling errors, like "Excell" in the abstract. 2. Bibliometric analysis does not require excessive emphasis and the introduction should clearly state the study's objectives and significance. 3. When discussing trends in literature production, ensure a detailed explanation of the data, including key events or trend drivers for each time period. 4. Provide a more detailed interpretation of the timeline analysis results, especially how the evolution of different research themes and methodologies is explained within the historical context, and consider comparing with findings from other studies. 5. Provide a clear description of the clustering analysis method and results, including the main keywords for each thematic cluster and citations of the most representative literature. some recently studies could be added in the discussion, such as ‘Effects of Fatigue in Lower Back Muscles on Basketball Jump Shots and Landings’, Physical Activity and Health, 6(1), p. 273–286. 6. Ensure the conclusion section provides a comprehensive summary of the overall research findings and highlights the study's significance for future research and clinical practice. Consider offering more specific future research suggestions to help readers further explore the application value of this study's findings. Comments on the Quality of English Language

Moderate editing of English language required

Author Response

Thank you for reading, reviewing and editing our article. Your reviews will contribute to a significant improvement in the quality of the article. We took the comments into account and entered them into the article (marked in yellow). At the same time, we thank you for your compliments and we are glad to have the opportunity to publish our article in your magazine.

  1. The English should be fluent and clear, with smooth transitions between paragraphs. Correct spelling errors, like "Excell" in the abstract.

Text partially has been rearranged (pages 1-15).

  1. Bibliometric analysis does not require excessive emphasis and the introduction should clearly state the study's objectives and significance.

The introduction has been rewritten to emphasize the study aims and objectives. The more detailed description of bibliometrics has been removed from the Introduction and is fully explained in the Methodology section.

  1. When discussing trends in literature production, ensure a detailed explanation of the data, including key events or trend drivers for each time period.

Detailed interpretation is explained in the added chapter Discussion (pages 13-15).

  1. Provide a more detailed interpretation of the timeline analysis results, especially how the evolution of different research themes and methodologies is explained within the historical context, and consider comparing with findings from other studies.

Detailed interpretation is explained in the added chapter Discussion (pages 13-15).

  1. Provide a clear description of the clustering analysis method and results, including the main keywords for each thematic cluster and citations of the most representative literature. some recently studies could be added in the discussion, such as ‘Effects of Fatigue in Lower Back Muscles on Basketball Jump Shots and Landings’, Physical Activity and Health, 6(1), p. 273–286.

The methodology section was extended with the SKS methodology (pages 2-3) which was used to analyse author keywords landscape and their clusters. The representative  author keywords for each cluster used in the analysis are represented and the most important papers are shown in Table 3 (pages 8-11), which has been restructured. The suggested reference was added to References.

  1. Ensure the conclusion section provides a comprehensive summary of the overall research findings and highlights the study's significance for future research and clinical practice. Consider offering more specific future research suggestions to help readers further explore the application value of this study's findings.

Detailed interpretation is explained in the added chapter Discussion (pages 13-15).

Reviewer 2 Report

Comments and Suggestions for Authors

excellent research, perhaps add a map with paper distribution

Author Response

Thank you for reading, reviewing and editing our article. Your reviews will contribute to a significant improvement in the quality of the article. We took the comments into account and entered them into the article (marked in yellow). At the same time, we thank you for your compliments and we are glad to have the opportunity to publish our article in your journal.

Excellent research, perhaps add a map with paper distribution.

The paper distribution is already shown in Table 3 (pages 8-11).

Reviewer 3 Report

Comments and Suggestions for Authors

Thank you for the opportunity to review this article. It is well written and structured. Its brevity also makes it easy to read.

The aim of the study is to comprehensively analyze studies and identify the most prolific research organizations and research topics related to physiotherapy and chronic low back pain (CLBP) using thematic, chronological analysis and bibliometric approach.

The major contribution: The manuscript uses an original mixed-method approach to explore the most popular research themes. Thus, authors provided an interdisciplinary and comprehensive view of the research field. 

Value/Importance to the readers/science: Bibliometric study contributes to the further research in this field providing multidisciplinary knowledge about CLBP. On the other hand, chronological analysis helps to identify prospective research trends. 

However, there are some issues to be addressed.

References starting from 87 to 107 are not cited anywhere.

The authors state: "Newer research since 2018 mostly refers to cohort analyses using newer clinical assessment tools (such as Numerical Pain Rating System, Oswestry disability index, Rolland-Morris questionnaire) [84–86] with the aim of developing the best possible clinical outcome of CLBP rehabilitation and treatment." Some new studies were not mentioned here. High-intensity laser therapy is also a type of physiotherapy that had good results in all aspects (pain intensity, Oswestry disability index and Roland disability index). For example, systematic reviews with meta-analyses on high-intensity laser therapy in low back pain management could be mentioned either here or in Table 3.

Minor issues:

Abstract:

Probably it should be “quantitatively and qualitatively” in row 13. Also a dot could be mistakenly placed here in the middle of the sentence in row 14. 

Remove the dot in the middle of the sentence:

"The article quantitatively and quantitatively analyzed research literature production harvested from the Scopus bibliometric database. using a triangulation of bibliometric 14 and thematic analysis."

Table 1:

Should 'informations' be single here? Please double check here "Percentage of funded informations sources (in %)"

Figure 2:

If you navigate pointer over the figure, some text appears, e.g. "Slika, ki vsebuje besede ...", please remove it. Please check this issue in other Figures as well.

Author Response

Thank you for reading, reviewing and editing our article. Your reviews will contribute to a significant improvement in the quality of the article. We took the comments into account and entered them into the article (marked in yellow). At the same time, we thank you for your compliments and we are glad to have the opportunity to publish our article in your journal.

References starting from 87 to 107 are not cited anywhere.

References has been rearranged and added in accordance with text change.

The authors state: "Newer research since 2018 mostly refers to cohort analyses using newer clinical assessment tools (such as Numerical Pain Rating System, Oswestry disability index, Rolland-Morris questionnaire) [84–86] with the aim of developing the best possible clinical outcome of CLBP rehabilitation and treatment." Some new studies were not mentioned here. High-intensity laser therapy is also a type of physiotherapy that had good results in all aspects (pain intensity, Oswestry disability index and Roland disability index). For example, systematic reviews with meta-analyses on high-intensity laser therapy in low back pain management could be mentioned either here or in Table 3.

More recent studies of CLBP laser therapy has been added (page 12).

Minor issues:

Abstract:

Probably it should be “quantitatively and qualitatively” in row 13. Also a dot could be mistakenly placed here in the middle of the sentence in row 14. 

Corrected.

Remove the dot in the middle of the sentence:

Corrected.

"The article quantitatively and quantitatively analyzed research literature production harvested from the Scopus bibliometric database. using a triangulation of bibliometric 14 and thematic analysis."

Corrected.

Table 1:

Should 'informations' be single here? Please double check here "Percentage of funded informations sources (in %)"

Corrected.

Figure 2:

If you navigate pointer over the figure, some text appears, e.g. "Slika, ki vsebuje besede ...", please remove it. Please check this issue in other Figures as well.

Corrected.

Reviewer 4 Report

Comments and Suggestions for Authors

In general, the article titled "Trends in Physiotherapy of Chronic Low Back Pain Research: Knowledge Synthesis Based on Bibliometric Analysis" is very well executed and of interest to the scientific community. Here are some detailed suggestions for improvement, although in my opinion, it should be accepted for publication:

1. Abstract:

   - The abstract is comprehensive but could be more concise. Consider summarizing the key findings more succinctly to enhance readability.

   - Specify the main themes identified in the thematic analysis directly in the abstract.

2. Introduction:

   - The introduction provides a good overview but could benefit from a clearer statement of the research gap. Explain why this bibliometric analysis is necessary compared to previous reviews.

3. Language and Grammar:

   - Ensure consistent use of tense and avoid passive voice where possible. For example, "The article quantitatively and quantitatively analyzed" should be revised for clarity.

   - Avoid jargon and complex sentences that may confuse readers unfamiliar with the topic.

Methodology

1. - The methodology section is well-detailed but could be better organized for clarity. Consider dividing it into subsections (e.g., Data Collection, Data Analysis).

Results

1.   - Use more visual aids like graphs and tables to present the data. For example, a line graph showing the exponential growth trend after 2018 would be beneficial.

2. - Provide a clearer explanation of how themes and clusters were identified and named. Include examples to illustrate the process.

   - The six identified clusters should be briefly described in the results section before being elaborated upon in the discussion.

Discussion

1.- The discussion should delve deeper into the implications of the findings. How do these trends impact future research or clinical practice in physiotherapy for CLBP?

   - Discuss any surprising findings and potential reasons behind them.

   - Compare the results with findings from previous bibliometric studies to contextualize the study's contributions.

3. Limitations:   - Acknowledge the limitations of the study, such as potential biases in the database.

 By addressing these points, the article can become more coherent, impactful, and valuable to the scientific community studying physiotherapy and chronic low back pain.

Author Response

Thank you for reading, reviewing and editing our article. Your reviews will contribute to a significant improvement in the quality of the article. We took the comments into account and entered them into the article (marked in yellow). At the same time, we thank you for your compliments and we are glad to have the opportunity to publish our article in your journal.

In general, the article titled "Trends in Physiotherapy of Chronic Low Back Pain Research: Knowledge Synthesis Based on Bibliometric Analysis" is very well executed and of interest to the scientific community. Here are some detailed suggestions for improvement, although in my opinion, it should be accepted for publication:

  1. Abstract:

   - The abstract is comprehensive but could be more concise. Consider summarizing the key findings more succinctly to enhance readability.

The abstract has been supplemented.

   - Specify the main themes identified in the thematic analysis directly in the abstract.

 The main themes of the thematic analysis has been specified in the abstract.

  1. Introduction:

   - The introduction provides a good overview but could benefit from a clearer statement of the research gap. Explain why this bibliometric analysis is necessary compared to previous reviews.

The research gap has been presented in more detail (page 2).

  1. Language and Grammar:

   - Ensure consistent use of tense and avoid passive voice where possible. For example, "The article quantitatively and quantitatively analyzed" should be revised for clarity.

   - Avoid jargon and complex sentences that may confuse readers unfamiliar with the topic.

The text has been partially rearranged.

Methodology

  1. - The methodology section is well-detailed but could be better organized for clarity. Consider dividing it into subsections (e.g., Data Collection, Data Analysis).

 The methodology section has been rewritten and split into  2,3 sub sections (pages 2-3).

Results

  1. - Use more visual aids like graphs and tables to present the data. For example, a line graph showing the exponential growth trend after 2018 would be beneficial.

A graph with a visual representation of the growth trend from 2018 onwards has been added (page 6).

  1. - Provide a clearer explanation of how themes and clusters were identified and named. Include examples to illustrate the process.

Due to the fact, that the interpretation of the blibliometrics method is purely subjective, we as researchers searched for the connections between various themes/keywords and according to our professional knowledge on the specific research field we identified and consequently named clusters. This explanation was added in the text (pages6-8).

   - The six identified clusters should be briefly described in the results section before being elaborated upon in the discussion.

The main clusters of the thematic analysis are briefly described in results (pages 6).

Discussion

1.- The discussion should delve deeper into the implications of the findings. How do these trends impact future research or clinical practice in physiotherapy for CLBP?

   - Discuss any surprising findings and potential reasons behind them.

   - Compare the results with findings from previous bibliometric studies to contextualize the study's contributions.

An extensive discussion has been written with an explanation of the findings (pages 13-15).

  1. Limitations: - Acknowledge the limitations of the study, such as potential biases in the database.

The study limitation section has been added (page 13).

By addressing these points, the article can become more coherent, impactful, and valuable to the scientific community studying physiotherapy and chronic low back pain.

Reviewer 5 Report

Comments and Suggestions for Authors

General Comments

The manuscript "Trends in Physiotherapy of Chronic Low Back Pain Research: Knowledge Synthesis Based on Bibliometric Analysis" provides a comprehensive bibliometric analysis of research trends in physiotherapy for chronic low back pain (CLBP). The authors have successfully identified key themes, prolific research entities, and chronological developments in the field. This study is a valuable contribution to the existing literature, offering insights into the evolution of research topics and helping to identify future research directions.

Strengths

Comprehensive Data Collection: The use of the Scopus database to harvest 2843 data sources spanning from 1974 to 2024 ensures a broad and representative sample of the literature.

Methodological Rigor: The application of both quantitative and qualitative bibliometric methods, including the use of VOSviewer for thematic mapping, provides a robust framework for analysis.

Clear Research Questions: The study is well-structured around three clear research questions that guide the analysis effectively.

Visual Representations: The inclusion of scientific landscapes and chronological diagrams aids in visualizing the complex relationships and developments within the research field.

Specific Comments

Abstract and Introduction:

The abstract effectively summarizes the study, but it could be enhanced by briefly mentioning the implications of the findings for future research and practice.

The introduction provides a good background but could benefit from a more detailed discussion on the significance of bibliometric analysis in advancing the understanding of CLBP research trends.

Methods:

The search strategy is well-documented. However, additional details on the criteria for inclusion and exclusion of sources would enhance the reproducibility of the study.

The rationale for using the Scopus database is clear, but discussing the potential limitations of relying on a single database would be beneficial.

Results:

The results are comprehensive and well-organized. The identification of six main themes and the chronological analysis are particularly noteworthy.

The presentation of data in tables and figures is clear, but some tables (e.g., Table 2 on prolific journals) could benefit from more detailed footnotes explaining the abbreviations and metrics used.

Discussion and Conclusion:

The discussion effectively interprets the findings, but it could be strengthened by integrating more recent literature to contextualize the results.

The conclusion succinctly summarizes the study's contributions but should also highlight specific recommendations for future research directions based on the identified trends.

Technical Aspects:

There are minor typographical and grammatical errors throughout the manuscript that need correction for clarity and readability.

Author Response

Thank you for reading, reviewing and editing our article. Your reviews will contribute to a significant improvement in the quality of the article. We took the comments into account and entered them into the article (marked in yellow). At the same time, we thank you for your compliments and we are glad to have the opportunity to publish our article in your journal.

Abstract and Introduction:

The abstract effectively summarizes the study, but it could be enhanced by briefly mentioning the implications of the findings for future research and practice.

Text in abstract has been added.

The introduction provides a good background but could benefit from a more detailed discussion on the significance of bibliometric analysis in advancing the understanding of CLBP research trends.

The introduction has been rewriten, to present  the benefits of the bibliometric analysis (pages 1-2-).

Methods:

The search strategy is well-documented. However, additional details on the criteria for inclusion and exclusion of sources would enhance the reproducibility of the study.

No further exclusion or inclusion criteria were used. That statement wss also added to the Section search procedure (page 2).

The rationale for using the Scopus database is clear, but discussing the potential limitations of relying on a single database would be beneficial.

The study limitation section has been added (page 13).

Results:

The results are comprehensive and well-organized. The identification of six main themes and the chronological analysis are particularly noteworthy.

The main themes of the thematic and chronological analysis has been specified in the results (pages 6-11).

The presentation of data in tables and figures is clear, but some tables (e.g., Table 2 on prolific journals) could benefit from more detailed footnotes explaining the abbreviations and metrics used.

Footnotes were added (page 4).

Discussion and Conclusion:

The discussion effectively interprets the findings, but it could be strengthened by integrating more recent literature to contextualize the results.

Detailed interpretation is explained in the added chapter Discussion (page 13-15).

The conclusion succinctly summarizes the study's contributions but should also highlight specific recommendations for future research directions based on the identified trends.

Detailed interpretation is explained in the added chapter Discussion (page 13-15).

Technical Aspects:

There are minor typographical and grammatical errors throughout the manuscript that need correction for clarity and readability.

The text has been partially rearranged.

Round 2

Reviewer 3 Report

Comments and Suggestions for Authors

Overall, the authors did a very good job, but a couple issues were not fixed.

Abstract: a dot could be mistakenly placed here in the middle of the sentence in row 14. 

Figure 2: If you navigate the pointer over the figure, some text appears, e.g. "Slika, ki vsebuje besede ...", which should be removed, I think.

Author Response

The comments have been taken into account.